# A Place-Based Conceptual Model (PBCM) of *Neotricula aperta/Schistosoma mekongi* habitat before and after dam construction in the Lower Mekong River

**Guy R. Lanza**[1]*, **Suchart Upatham**[2], **Ang Chen**[3]

**1** Division of Environmental Sciences, College of Environmental Sciences and Forestry (ESF), State University of New York, Syracuse, New York, United States of America, **2** Department of Biology, Faculty of Science, Mahidol University, Bangkok, Thailand, **3** China Three Gorges Corporation, Wuhan, China

* glanza40@gmail.com

**Data Availability Statement:** All relevant data are within the manuscript.

## Abstract

In 1971, scientists from Mahidol University in Thailand and the Smithsonian Institution in the USA formed a research team to study a new species of *Schistosoma* in the Mekong River in Thailand and Laos. The studies, completed during 1971–1973, prior to the construction of any dams or restrictions to the natural flow regime of the Mekong River, provide a unique description of the natural ecological state of the river that can serve as a baseline for current research. The natural transmission of *Schistosoma japonicum*, Mekong Strain, was first reported on Khong Island, Laos in 1973 using sentinel mice. The first detailed description of the habitat ecology of the snail vector *Neotricula aperta* was done on-site in 1971 simultaneously with that research and is unique in providing the only description of the river shoreline habitat before any dams were built and any alteration of the natural flow regime was in place. Aggregating current information in a Place-Based Conceptual Model (PBCM) as an organizing template, along with current habitat models that combine ecological data with e-flows, can be developed and used as a tool to predict suitable habitats for snails. The natural flow regime of the Mekong River prior to any impoundments is described with current updates on the potential impacts of climate change and dams with flow-related snail habitat characteristics, including sediment drift and water quality. The application of the PBCM to describe and compare descriptive information on current and potential future *N. aperta/S. mekongi* habitat is discussed.

## Author summary

We provide a unique profile of *Neotricula aperta* habitat in the Mekong River in Laos and Thailand during the transmission of *Schistosoma mekongi* prior to the construction of any dams on the river. A Place-Based Conceptual Model (PBCM) is described to help identify potential new areas, with suitable habitats for snails and disease transmission. The natural flow regime of the Mekong River prior to any impoundments is described with current

**Funding:** The author(s) received no specific funding for this work.

**Competing interests:** The authors have declared that no competing interests exist.

updates on the potential impacts of climate change and flow-related snail habitat characteristics, including sediment drift and water quality. Environmental management strategies are outlined, emphasizing environmental flow (e-flow) and habitat models, with the potential to mitigate *N. aperta* distribution and S. mekongi disease transmission. The PBCM can serve as a template to aggregate and compare historical baseline information (Before dams) with current data (After dams) to create predictive models describing suitable habitats for *N. aperta* and other snail intermediate hosts and vectors along shoreline areas of the Mekong River continuum. The PBCM model could be a powerful management tool when combined with habitat suitability and e-flow models. Adding schistosome biomarkers currently under development in a One Health context to select possible areas of a river likely to support disease transmission could extend the value of the model.

## Introduction

We describe a Place-Based Conceptual Model (PBCM) using pre-and post-dam data from the lower Mekong River in Thailand and Laos. The PBCM describes snail habitats and can be an effective tool to identify potential new foci for *Neotricula aperta* and *Schistosoma mekongi* along the Mekong River as shown in (Fig 1). The Mekong River continuum is approximately 4,909 km long with an average annual discharge of 14,500 $m^3$/s [1]. The lower Mekong River basin is a key biodiversity hotspot with an invertebrate species array dominated by gastropods. Up-to-date information on shoreline habitat and snail communities is needed to help predict potential disease transmission sites and to conserve invertebrate biodiversity [2, 3]. In 1971 scientists from Mahidol University in Thailand and the Smithsonian Institution in the USA formed a research team to study a suspected new species of *Schistosoma japonicum* in the Mekong River in Thailand and Laos. In 1973 the team published the first report of the natural transmission of *Schistosoma japonicum*, Mekong Strain, on Khong Island, Laos (Latitude $14^0 7^/ 30^{//}$N, Longitude $105^0 51^/ 45^{//}$E) using sentinel mice [4]. These studies also described the habitat ecology of *N. aperta* during the transmission of *S. mekongi* in Thailand and Laos before any dams were built on the river, including data on the community, of snails, plants, algae, bacteria, and water and sediment quality [4,5]. The research provides a unique baseline pre-impoundment (Before Dams) description of the biota, physical parameters, and river water quality. This report provides a Place-Based Conceptual Model (PBCM) that compares information describing the habitat of *N. aperta* during the transmission of S. *mekongi* before dams were built on the Mekong River with the results from more recent post-dam studies of the Lower Mekong River in Laos, Thailand, and Cambodia [5,6]. The model uses current reports describing climate change and other factors that can cause impacts and altered river flow/sediment transport caused by dams to predict potential future *N. aperta* habitat and suggests management strategies to reduce disease transmission.

Climate change can alter the habitat for snails described in the PBCM by driving changes in weather patterns, flow regimes, water quality, sediment transfer, and other key environmental factors. Wang [7] notes that the Lower Mekong River Basin (LMRB) faces two major challenges to its water resources in the 21st century: potential climate change and ongoing and planned dam constructions. Alla [8] reviewed the impacts of dams and noted that they result in major negative impacts on water quality, aquatic ecology, land, terrestrial wildlife, vegetation, and air quality and recommended mitigation measures that should be put in place to reduce these negative impacts on the environment. A changing climate could modify rainfall patterns leading to changes in the hydrological cycle, including shifts in monsoon patterns

| Before Dams | After Dams | Future |
|---|---|---|
| Historical Habitat Descriptions 1971-1973<br><br>1. Shoreline (littoral)<br>2. Offshore (channel current) | Selected Updated Descriptions 2008-2022<br><br>1. Shoreline (littoral)<br>2. Offshore (channel current) | Selection of Potential Habitat Sites for More Detailed Study Using PBCM and Recent Habitat Descriptions and Models |
| Sediment, Water Quality, Biota Figures 1-5 Tables 1 & 2 | Sediment, Water Quality, Biota 9 sites in Laos, Thailand, Cambodia 2008-2010; Other Updates | Impacts of Climate Change and Human Activity |
| Habitat Influenced by Natural Flow Regime | Habitat Influenced by Impacts of Dams | Habitat Influenced by Management Strategies, E-flow applications |
| Monsoon Control | Regulated Flow of Water and Sediment from Dam Releases | Develop New Management Strategies |

**Fig 1. PBCM. A Placed-Based Conceptual Model based on pre- and post- dam construction conditions useful in predicting transmission of schistosomiasis resulting from future dam construction.**

that influence the water quality and sediments supporting the biota described in the PBCM. One report indicates that within the coming 20–30 years the operation of planned hydropower reservoirs is likely to have a larger impact on the Mekong hydrograph than the impacts of climate change, particularly during the dry season. The report also predicts that climate change will increase the uncertainty of the estimated reservoir operation impacts and that the direction of the flow-related changes induced by climate change is partly unclear [9]. When all the planned dam constructions are complete, the total active capacity of the LMRB is estimated to increase from 5 $km^3$ to more than 100 $km^3$. Laos lies almost entirely within the LMRB and its climate, landscape, and land use are the major factors shaping the hydrology of the river. The basin area of 202,000 $km^2$ represents 25% of the catchment area and 35% of the flow of the Mekong River Basin respectively [10]. The PBCM framework integrates ecological information describing the significance of the flood pulse and the natural flow regime [11,12], with the effects from dams and other barriers that change the natural flow, water quality [13], and

sediment distribution. Most of the fragmentation and disconnection caused by the world's 2.8 million dams is due to structures associated with reservoir areas greater than 1000 m$^2$ [14]. The PBCM framework has value in locating potential shoreline sites along the Mekong River that could provide habitat for snail intermediate hosts and vectors including *N. aperta*.

## Methods

Fig 2. Is a map showing the study sites in Ban Dan, Thailand, and Khong Island, Laos. On-site water chemistry included water quality and ion measurements over one monsoon cycle in Thailand and Laos (Figs 3–5). Diel cycles were completed on two dates during transmission of schistosomes with measurements of air temperature, water temperature, dissolved oxygen (DO), percent saturation and hydrogen ion concentration (pH) (Figs 6–9). Two stations were examined: one near shore with reduced current, extensive substrate, and snail communities and the other approximately 10 m offshore with swift current, very limited substrate, and no visible snail communities. Turbidity, total dissolved solids, electroconductivity, carbon dioxide, alkalinity, total hardness, calcium, magnesium, potassium, sodium, bicarbonate, and carbonate levels were measured at 2 diel intervals, 12 noon and midnight. A Model DR-EL portable water engineering kit (Hach Chemical Company) was used for on-site measurements. Additional chemistry was done in Bangkok on water and sediments by the water analysis laboratory of the Thai Department of Land Development, Ministry of Agriculture and Cooperatives following APHA procedures. Sediment analyses included particle size, organic carbon, available phosphorus, active iron, conductivity, soluble ions, and Cation Exchangeable Capacity (CEC). A 2-sample t-test for which the variances were not assumed to be equal was used for statistical analyses to compare mean values of 23 sediment parameters at Ban Dan, Thailand, and Khong Island, Laos (Fig 10. Biological samples were collected from natural and artificial substrates. Identification of aquatic plants, algae, and bacteria was done by the Biology Department at Khon Kaen University, and snail identification was done by the Faculty of Tropical Medicine and Hygiene at Mahidol University. Details of the methods used are provided in [3,4] (Figs 11 and 12).

The PBCM was built following methods developed by Brierley and Fryirs [15], and Sharma [16] using Before and After Dam habitat descriptions. *N. aperta* has also been reported as a host for *Paragonimus veocularis* and experimentally as a host for *P. heterotremus* [17].

The PBCM is presented in Fig 1, with historical 1971–1973 information (Before Dams) and updated information from 9 Lower Mekong River (LMR) sites in Laos, Thailand, and Cambodia surveyed by the Mekong River Commission during 2008–2010 (After Dams) [5,6]. The model includes site-specific changes after dams were built including habitat similarity and differences with pre-dam conditions regarding sediments, water quality, and biota.

## Results

### PBCM application

**Historical descriptions of *N. aperta* habitat (Before Dams 1971–73).** The historical information from the 1971–73 research (Before Dams) is provided in Figs 1–12.

The importance of water flow on the infection rate of *S. mekongi* was highly evident during pre-impoundment baseline studies. Although specific measurement of flow was not part of these studies, shoreline areas with relatively lower flow and frequent contact by humans and animals at Khong Island, Laos were associated with higher infection rates of *S. mekongi*. Offshore areas in deeper water with moving currents are relatively free of human contact and the mixing, dilution, and lack of substrate for attachment for *N. aperta* may explain the absence of significant infection (Figs 11–12). Upatham [18] studied the Mekong River in Thailand and

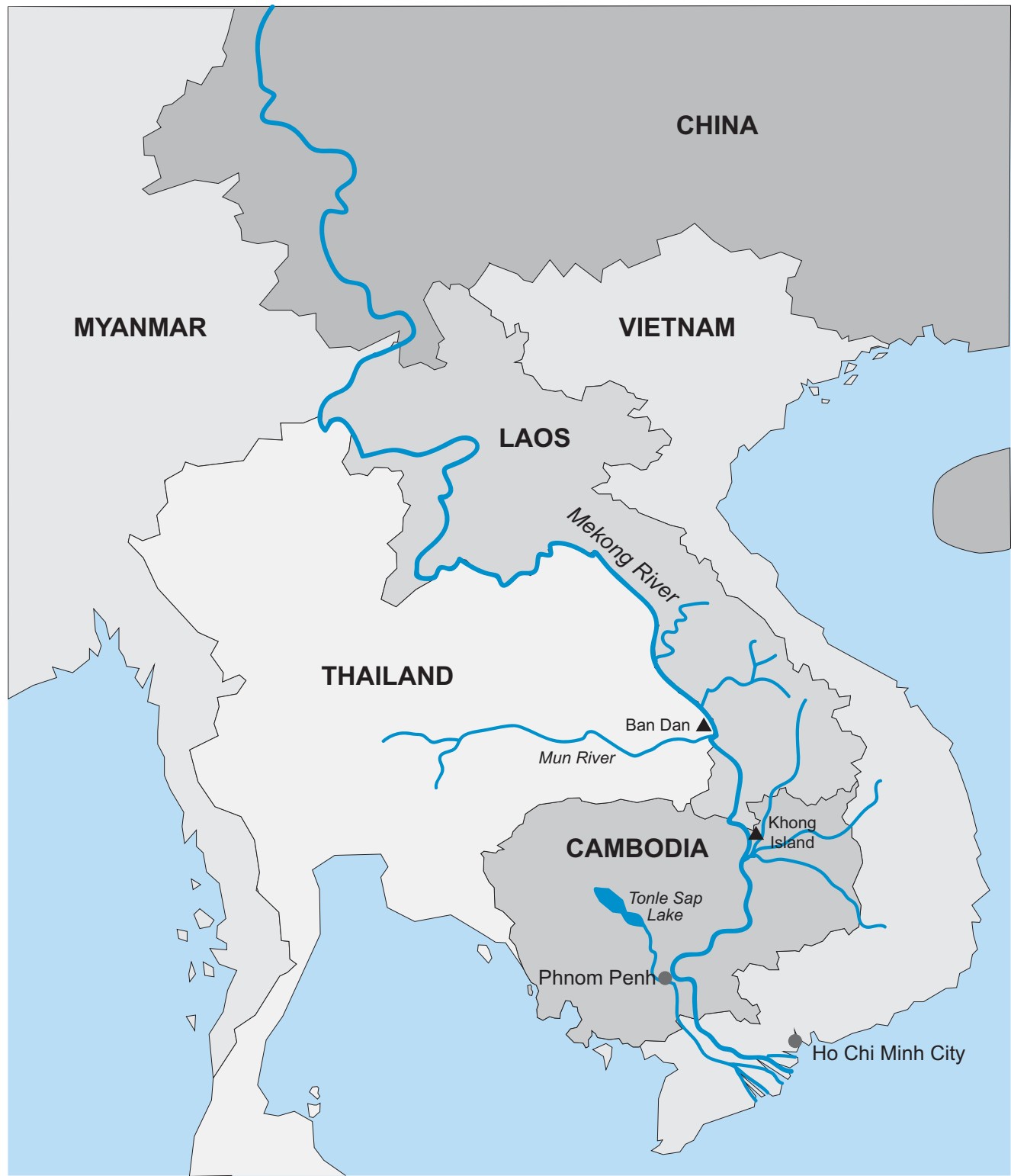

**Fig 2. Map of the study area** Map of Mekong River basin with study sites, Ban Dan, Thailand (MeB) and Khong Island, Laos (MeK).

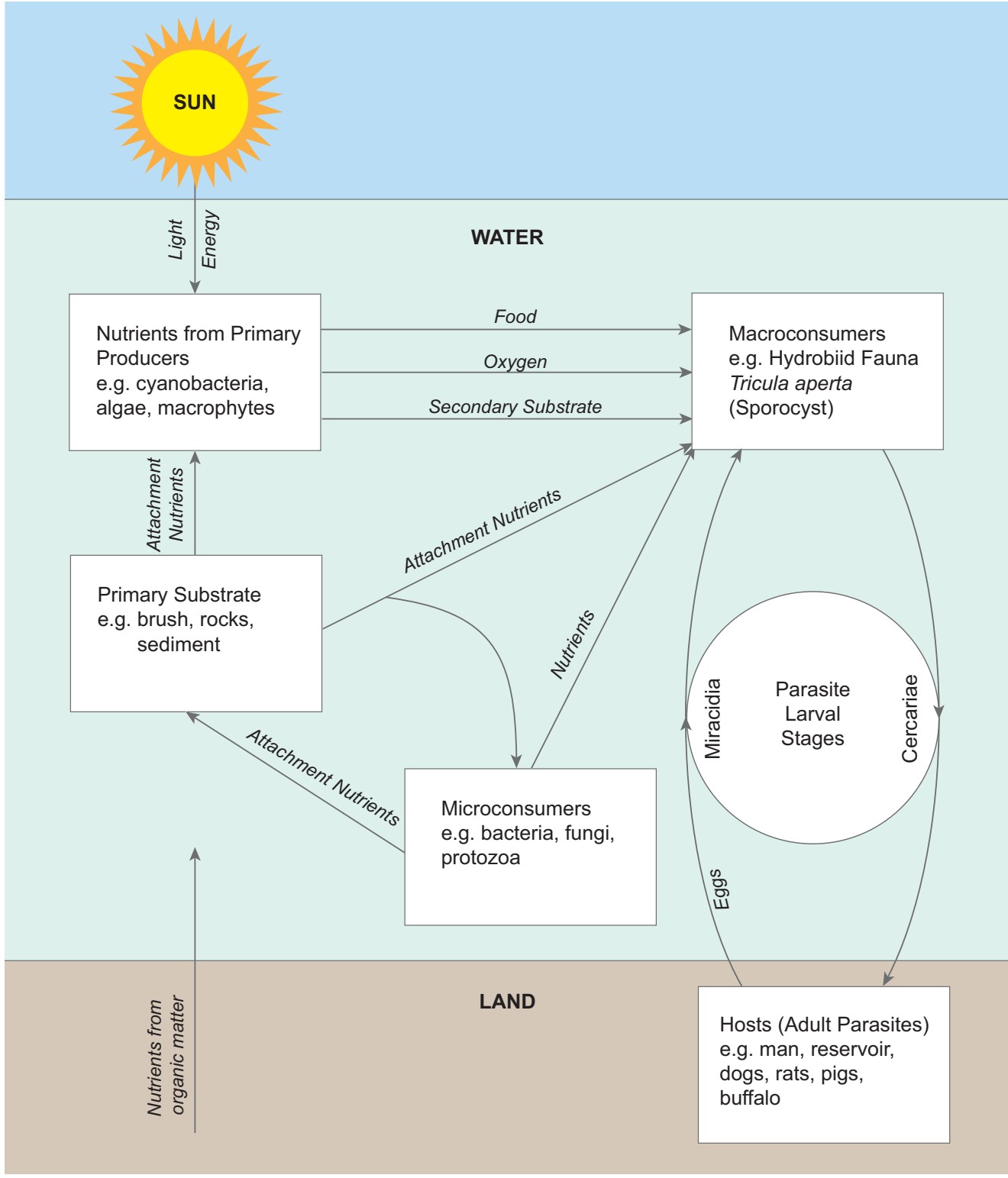

**Fig 3. Habitat description and food web relating aquatic and terrestrial nutrient cycles with snail and parasite life stages.**

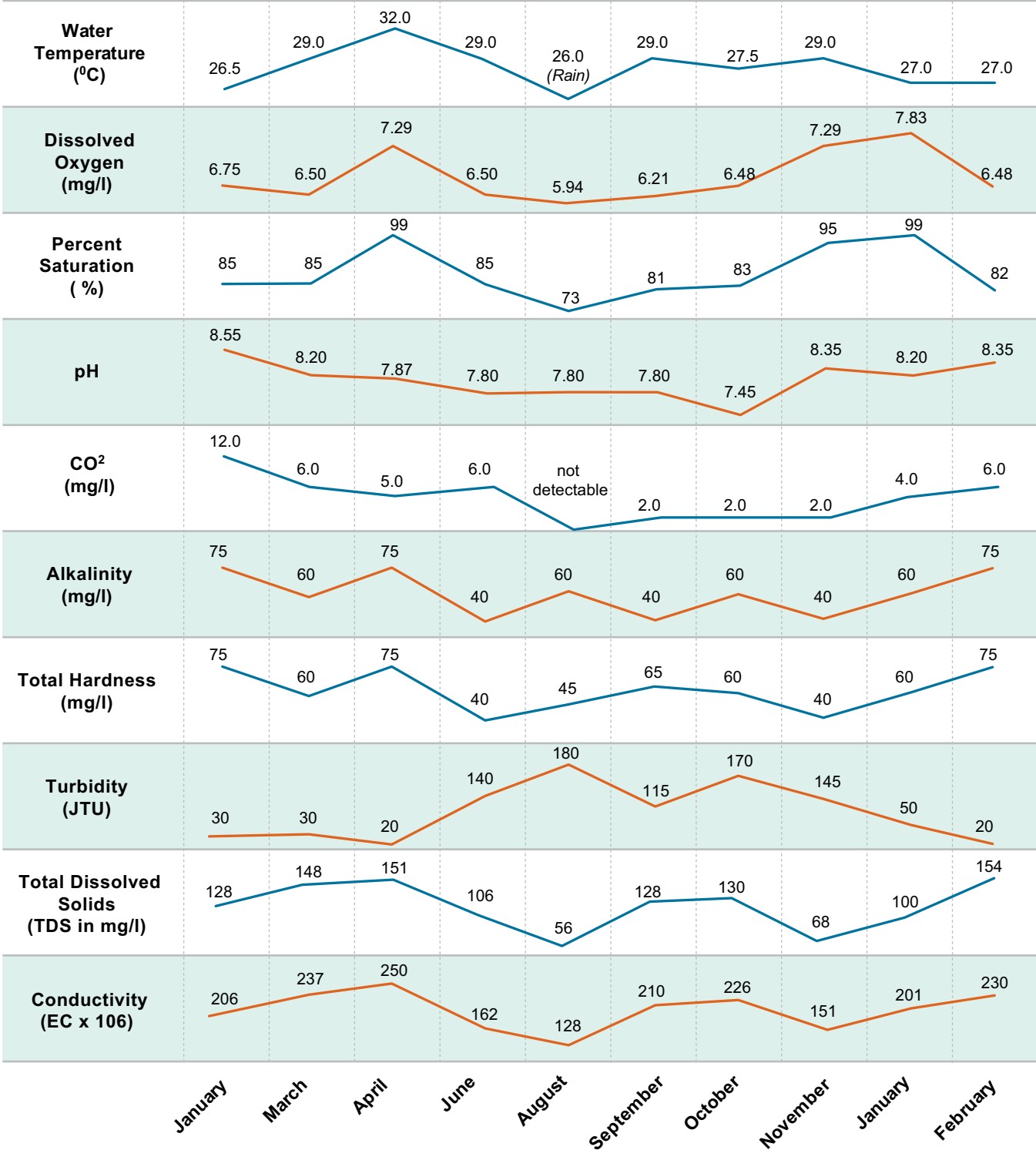

**Fig 4. Water quality one monsoon cycle.**

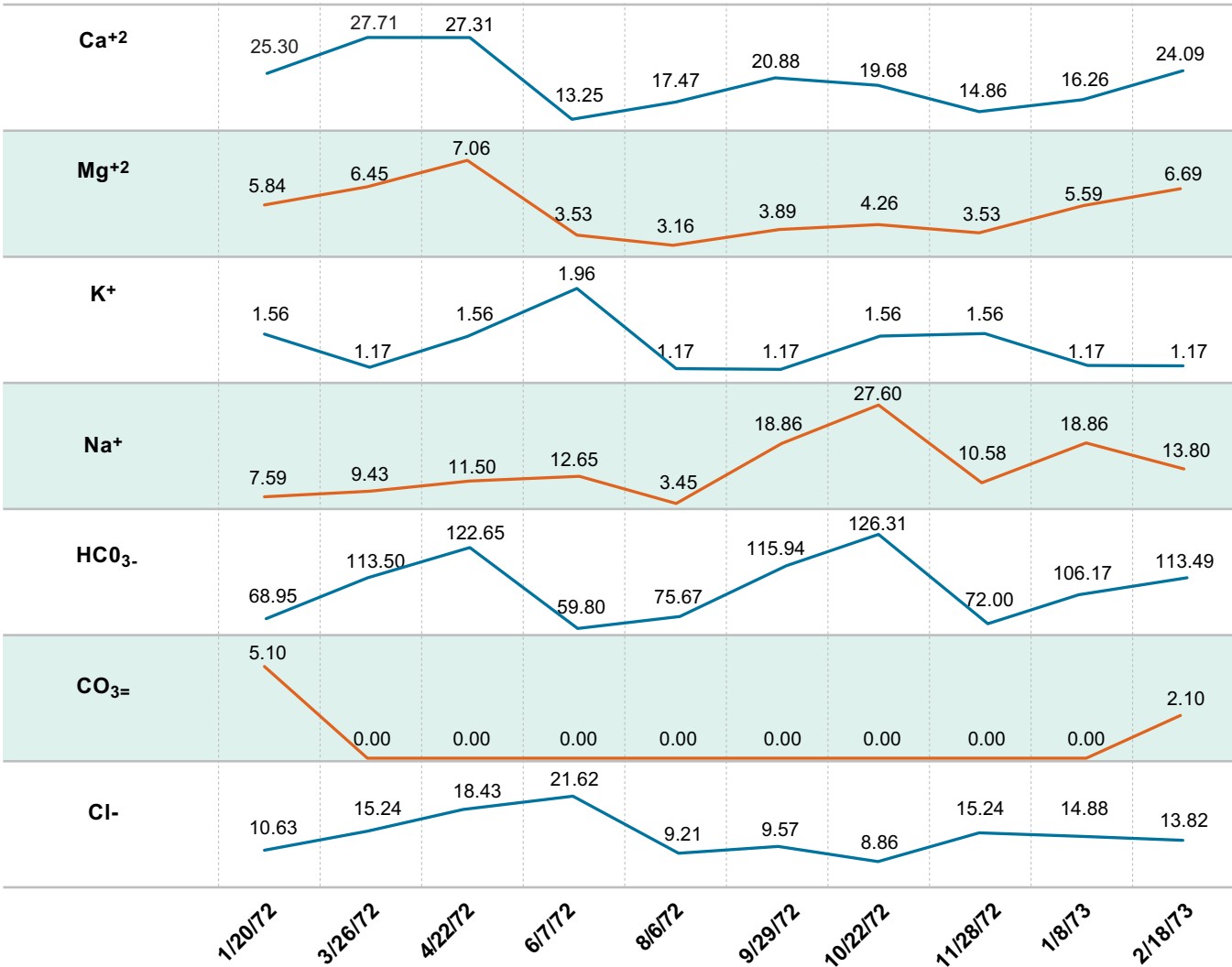

**Fig 5. Water ions one monsoon cycle.**

Laos and described the association of different water flow patterns on the bionomics of hydrobiid snails, including the alpha- and gamma-strains *N. aperta*. He reported higher densities of snails in areas with higher water flow, but data were not collected on the transmission of *S. mekongi* [18]. Site studies of favorable habitat for *N. aperta* at Khong Island prior to any changes in the natural flow regime of the Lower Mekong River indicate that the site reflected similar downstream water quality and sedimentation characteristics when compared to an upstream reference site at the village of Ban Dan, Thailand (Figs 2, 4, 5 and 10).

Water quality and water ion data from one monsoon cycle in the Mekong River prior to impoundment indicated a cyclic but stable habitat for snails, algae, and aquatic plants (Figs 4 and 5). Sediment data indicated lower levels of some parameters due to particles settling out with distance (Fig 10). Data from other studies of downstream stations in Vietnam showed similar trends in sediment transport and characteristics [19]. Statistically significant differences were noted in very fine sand, organic carbon, active iron, and $Ca^{2+}$ (Fig 10. Studies of substrates favored by *N. aperta* indicate that they use hard substrates such as tree branches,

**April 21, 1972,  Location 1: Area near shoreline with little flow**

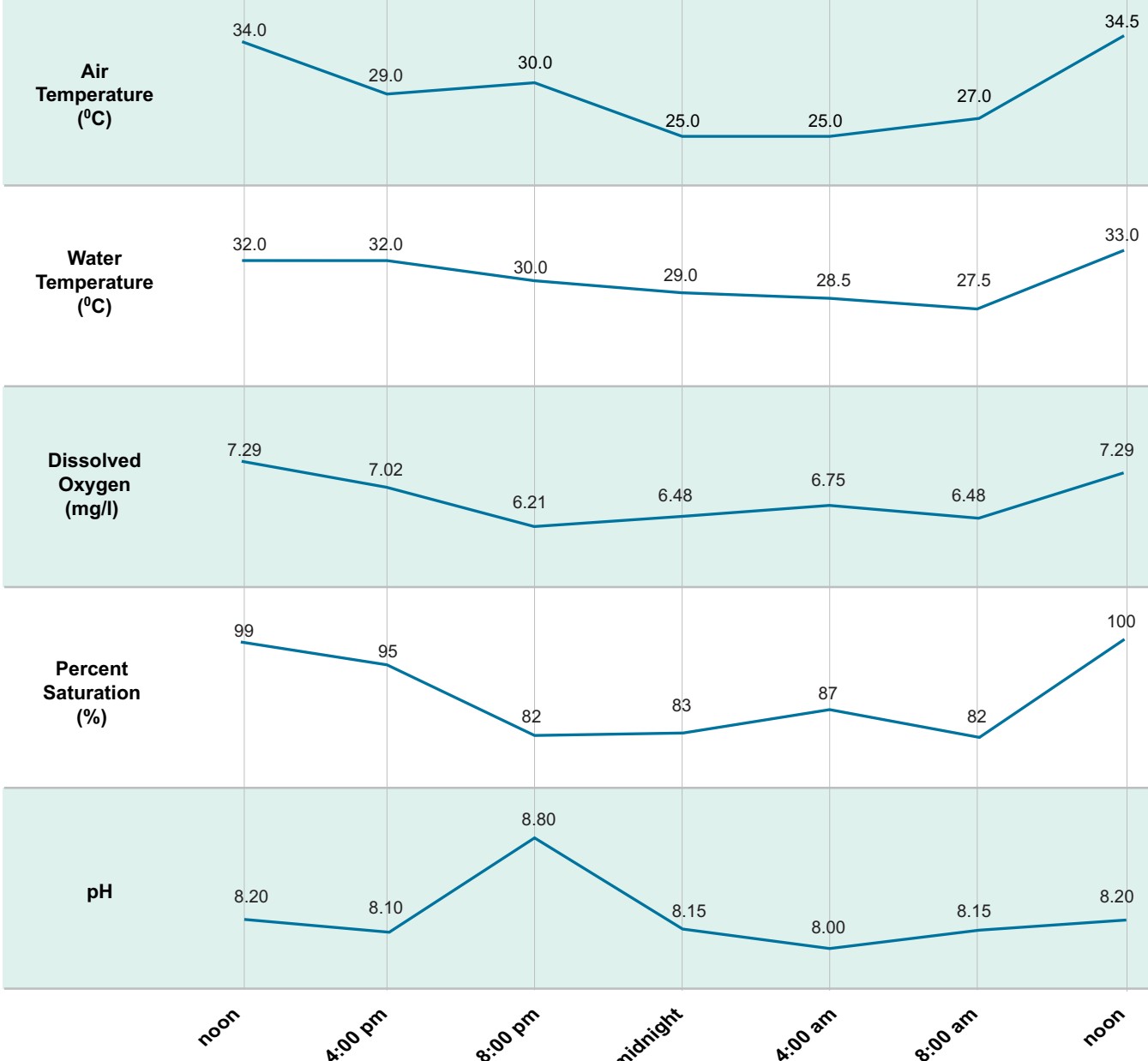

**Fig 6. 24-hour (diel) cycle April 21, 1972.**

leaves, metal, rocks, and aquatic plants [3] and that sand was seldom used by *N. aperta* (Figs 3, 9 and 10). Sediments are also a major source of nutrients supporting the primary producer community that provides substrate and food for *N. aperta* and other snail species of the Lower Mekong River. The microbial primary producer community on Khong Island had sixteen species that serve as food and substrate for *N. aperta* (Figs 11 and 12) and twenty-three species of dominant snails (Fig 12). The potential role of snail predation by fish, turtles, and other aquatic

**April 21, 1972, Location 2: Area offshore with noticeable flow**

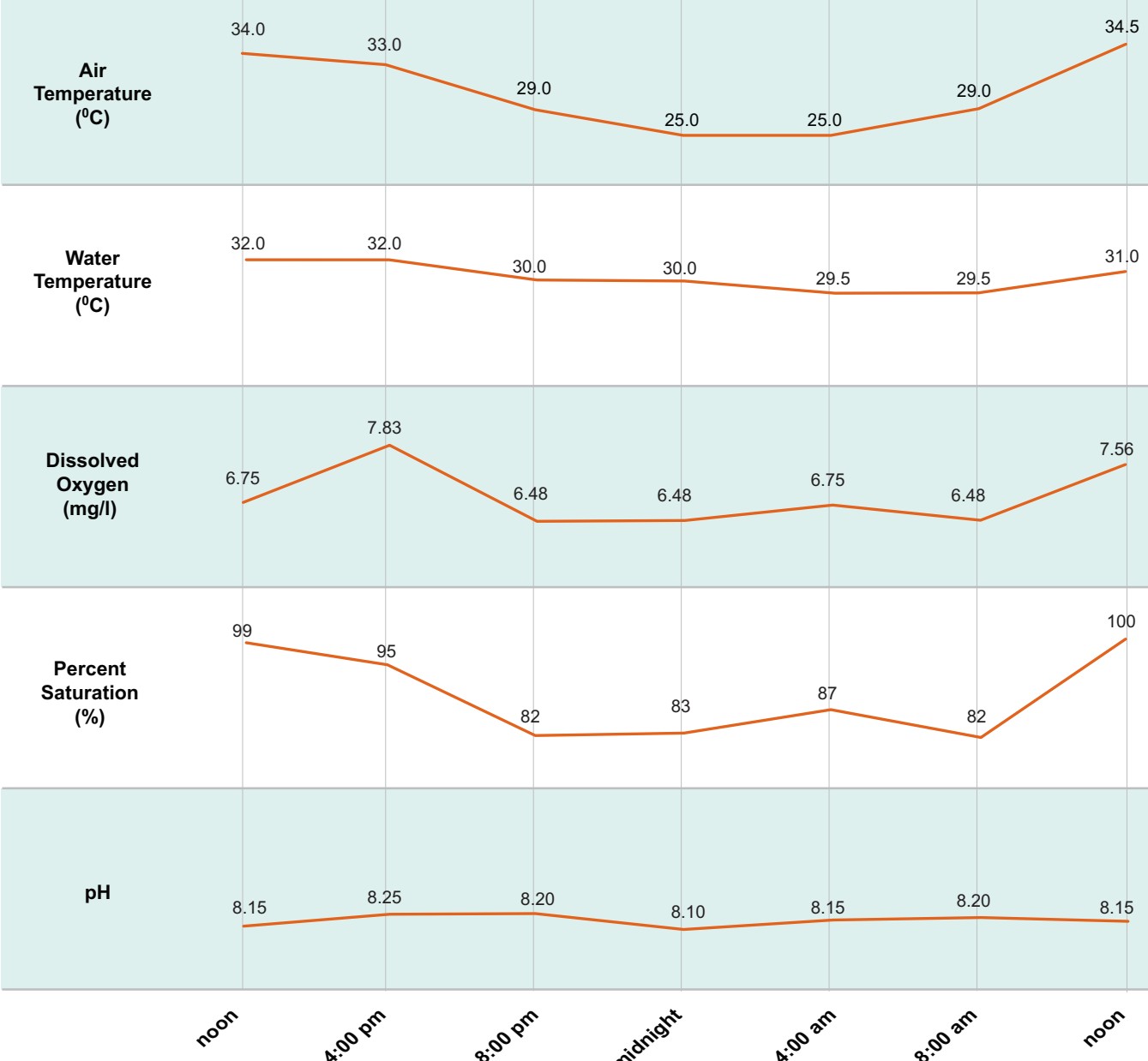

**Fig 7. 24-hour (diel) cycle April 21, 1972.**

organisms is a topic that needs study in the *N. aperta* habitat areas. A study of the use of fish to control parasitic diseases, including schistosomiasis in Lake Malawi, identified the challenges involved in biological control. [20] Mekong River fish that are snail predators include (*Yasuhi-kotakia morleti*) [21] and *Pangasius conchophilis*. [22] The snail-eating turtle *Malayemys sub-trijuga* in the Mekong River in Vietnam is known to be infected by 2 digenean species and could impact snail populations in different river areas. [23] The anticipated changes described

**April 27, 1972, Location 1: Area near shoreline with little flow**

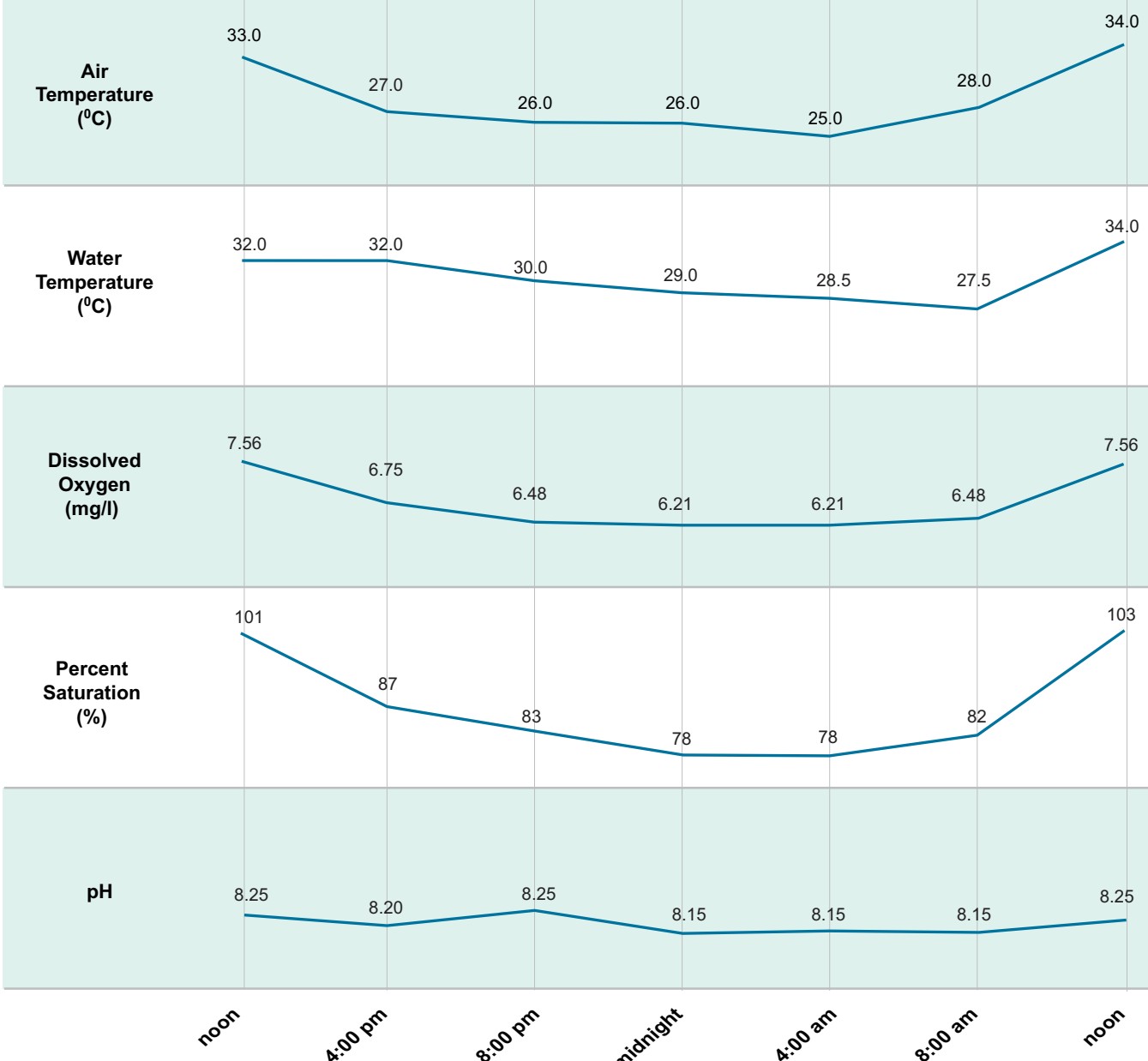

**Fig 8. 24-hour (diel) cycle April 27, 1972.**

in the updated *N. aperta* habitat descriptions could affect fish and turtle populations in the study area with impacts on snail predation.

**Current description of potential *N. aperta* habitat in the LMR (After Dam surveys) of 2008–2010.** The MRC survey data used in the PBCM identified several basic similarities and differences in the habitat comparisons between 1971–73 and 2008–2010 [5,6]. Our examples of applying the PBCM to predict potentially suitable habitats for *N. aperta* and other snails

**April 27, 1972, Location 2: Area offshore with noticeable flow**

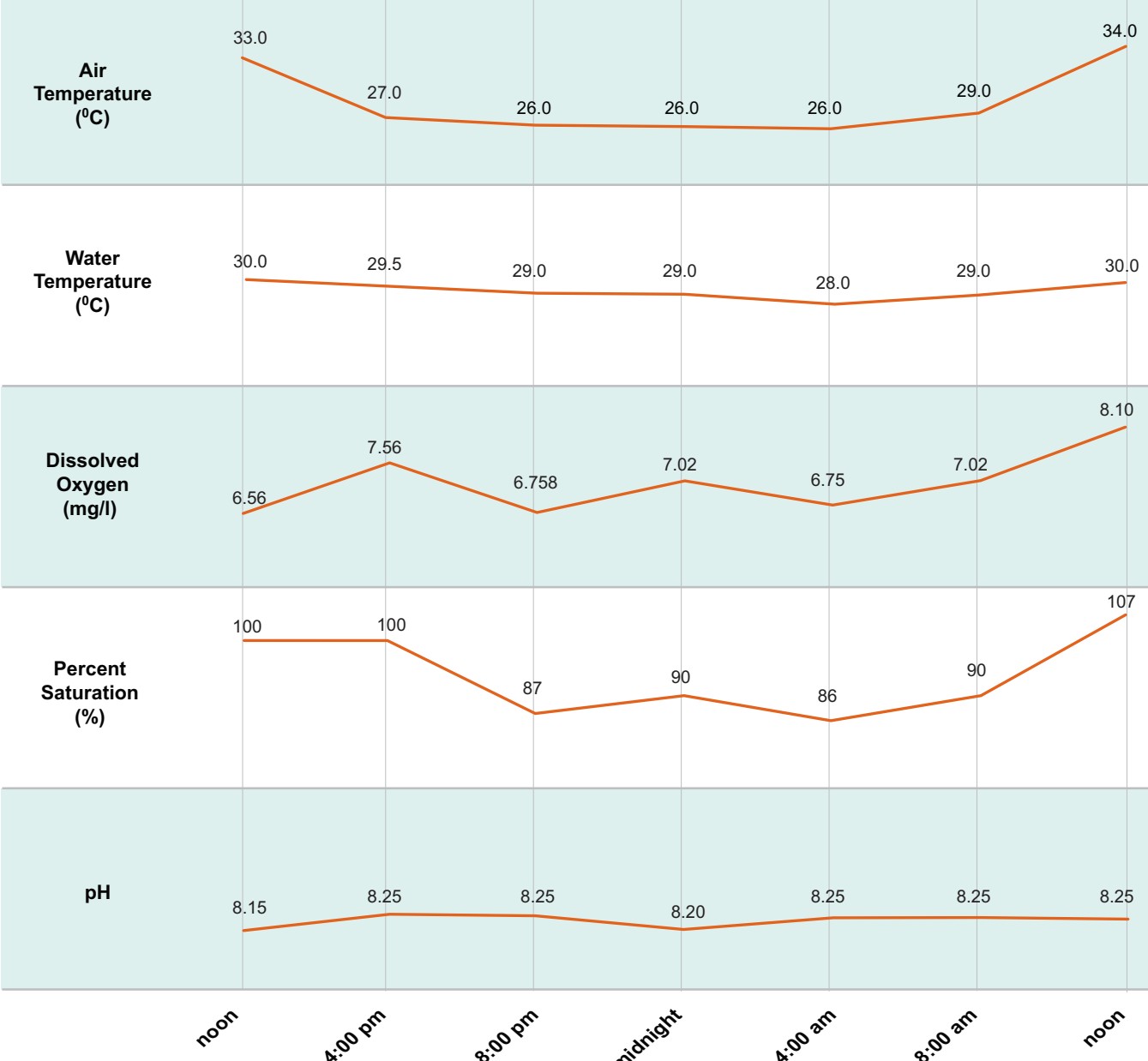

**Fig 9. 24 hour (diel) cycles April 27, 1972.**

compares data from recent surveys along with the historical baseline data from 1971–73. Recent reports of mainstream water quality were described as very good overall, but some areas with degraded habitat and water quality were reported. [5,6]. Although *N. aperta* was not reported from the surveys, one site surveyed at Done Ngiev, Laos was very similar to the Khong Island Laos transmission site with sand substrate, *Homomonia riparia* shrubs, and filamentous algae and diatoms. The PBCM identified the Done Ngiev site as a good match to

| Particle size (%)[a] | MeB[b] | | MeK[c] | | |
|---|---|---|---|---|---|
| | Mean | S.D. | Mean | S.D. | Significant |
| Very coarse sand (1.0-2.0) | 0.08 | 0.04 | 0.05 | 0.71 | |
| Coarse sand (0.05-1.0) | 0.53 | 0.20 | 1.33 | 1.81 | |
| Medium sand (0.25-0.05) | 3.20 | 1.95 | 1.34 | 0.85 | |
| Fine sand (0.10-0.25) | 29.25 | 18.80 | 25.41 | 10.70 | |
| Very fine sand (0.05-0.10) | 16.88 | 7.79 | 45.87 | 7.03 | x |
| Total sand (0.05-2.0) | 50.12 | 27.10 | 77.94 | 5.80 | |
| Silt (0.002-0.05) | 38.05 | 21.10 | 16.36 | 5.30 | |
| Clay (≤0.002) | 11.83 | 6.08 | 5.70 | 3.20 | |
| Organic carbon (%) | 0.75 | 0.33 | 0.25 | 0.09 | x |
| Available phosphorus (mg L$^{-1}$) | 30.43 | 3.81 | 31.2 | 3.82 | |
| Active iron (%) | 1.61 | 0.58 | 0.99 | 0.11 | x |
| Saturation extract ECe@25C (µmho) | 0.62 | 0.07 | 0.81 | 0.44 | |
| **Soluable cations (meq 100g$^{-1}$)** | | | | | |
| Ca$^{2+}$ | 4.93 | 0.63 | 5.80 | 2.43 | |
| Mg$^{2+}$ | 1.15 | 0.13 | 1.45 | 0.82 | |
| Na$^+$ | 0.76 | 0.13 | 1.21 | 0.89 | |
| K$^+$ | 0.13 | 0.02 | 0.33 | 0.38 | |
| **Soluable anions (meq 100g$^{-1}$)** | | | | | |
| Cl$^-$ | 0.60 | 0.16 | 1.00 | 0.52 | |
| HCO$_3^-$ | 3.87 | 0.45 | 3.39 | 0.44 | |
| CO$_3^-$ | 0.00 | 0.00 | 0.00 | 0.00 | |
| **Effective exchangeable cations (meq 100 g$^{-1}$)[d]** | | | | | |
| Ca$^{2+}$ | 13.63 | 6.20 | 7.12 | 1.12 | x |
| Mg$^{2+}$ | 1.28 | 0.75 | 0.86 | 0.25 | |
| Na$^+$ | 0.02 | 0.04 | 0.01 | 0.01 | |
| K$^+$ | 14.93 | 6.99 | 8.16 | 1.2 | |

a. Bracketed values are particle size ranges in mm

b. MeB, Mekong River at Ban Dan, Thailand *(n=6)*

c. MeK, Mekong River at Khong Island, Laos

d. Extractable cations (1 N KCl) minus soluable cations.

**Fig 10. Sediment characteristics Mekong River shoreline (MeB, MeK).**

| Microhabitat | Algal Assemblage | Gastropods on Algae |
|---|---|---|
| Algae attached to wood and rock outcroppings or as free-floating mats 5 meters off shore. | *Ulothrix* sp. with dense epiphytic communities of *Malleodendron* sp. and *Navicula* sp. | *Tricula aperta*<br>*Pachydrobia mcmulleni* |
| Free-floating macrophytes near shorelines. | *Azolla* sp. with epiphytic *Navicula* sp. | *Stenothyra hybocystoides*<br>*S. basisculpta* |
| Algae attached to submerged twigs: rocks and leaves near shoreline. | *Spirogyra* sp. *Oedogonium* sp. *Ulothrix* sp. with epiphytic *Malleodendron* sp. Twigs with epiphytic *Spirulina* sp. and *Fragilaria* sp. leaves with epiphytic Closterium sp. and *Navicula* sp. | *T. aperta*<br>*S. hybocystoids*<br>*S. basisculpta*<br>*S. cambodiensis*<br>*P. crooki*<br>*P. wykoffi*<br>*P. mcmulleni* |

**Fig 11. Algal-snail associations.**

Khong Island and would be a good site for more detailed follow-up habitat studies. Another similar site in Cambodia, at Kampi pool near the Lao border, had a sand substrate with some stones and a snail population of almost *400* Stenothyra sp., a common co-habitant of *N. aperta*. The PBCM also identified sites in Laos, Thailand, and Cambodia that are dissimilar to Khong Island and not good matches. Sites at Ban Xiengkok, Laos, Nakorn Panom, Thailand, and Se San, Cambodia had substrates dominated by rocks, gravel, and mud without much sand and generally low water quality and turbulent currents.

Shimada [24] described the high susceptibility of the gamma strain of *N. aperta to S. mekongi* from Khong Island, Laos. A 2017 survey from Khong Island noted the presence of more than 5500 *N. aperta* from the same shoreline areas studied in 1972, with some snails testing positive *for S. mekongi* [25]. More recently, Kumagai [26] developed a loop-mediated isothermal amplification (LAMP) assay for *S. mekongi* and evaluated the utility of the assay for detecting DNA of *S. mekongi* in snails and human stool and snail samples in endemic areas on Khon Island, Laos. The LAMP assay results were used to develop a risk map for monitoring *S. mekongi* and preventing epidemics. Although these surveys [25,26] did not include the habitat characteristics, the results strongly suggest that some of the favorable *N. aperta/S mekongi*

| Species | Locale** | | | | | | |
|---|---|---|---|---|---|---|---|
| | (1) | (2a) | (2b) | (3a) | (3b) | (4a) | (4b) |
| *Tricula aperta** | + | + | + | + | + | + | + |
| *Stenothyra basisculpta* Brant | + | + | + | + | - | + | - |
| *S. hybocystoides* Bavay | + | + | + | + | + | + | + |
| *S. laotiensis* Temcharoen | - | - | - | - | + | - | - |
| *S. cambodiensis* Brandt | - | + | - | - | - | - | - |
| *Stenothyra sp.* Juveniles | + | + | - | - | + | + | + |
| *Manningiella expansa* Brandt | + | + | - | - | + | + | + |
| *M. pellucida* Bavay | + | + | - | - | + | + | + |
| *Pachydrobia mcmulleni*[1] | - | + | - | - | - | - | - |
| *P. crooki* | - | + | - | - | - | - | - |
| *P. wykoffi* | - | + | - | - | - | - | - |
| *Pachydrobiella brevis* Bavay | + | + | - | - | + | + | + |
| *Jullienia acuta* Poirier | + | - | - | - | - | - | - |
| *J. harmandi* Poirier | + | - | - | - | - | - | - |
| *Paraprosothenia iijimai* Brandt | + | - | - | - | - | - | - |
| *P. levayi* Brandt | - | + | - | - | - | - | - |
| *Lacunopsis fisherpiettei* Brandt | - | + | - | - | - | - | + |
| *L. ventricose* Poirier | + | - | - | - | - | - | + |
| *Lacunopsis sp.* Juveniles | - | + | - | - | - | - | + |
| *Clea (Anentome) helena* Philippi | - | + | + | - | - | + | - |
| *Wattebledia crosseana* Wittebled | + | - | - | - | - | - | - |
| *Wattebledia sp.* Juveniles | - | - | - | - | - | + | - |
| *Hydrorissoia hospitalis* Brandt | + | - | - | - | - | - | - |
| *Bithynia sp.* Juveniles | - | - | - | - | - | - | + |

\*    *Lithoglyphopsis aperta* Temcharoen, vector of human Schistosomiasis

\*\*   Numbers in brackets identify east shore locale; a and b are stations collected within a locale.

+ and – represent present or absent respectively. See F-4 Kitikoon et al. 1973

(1) Military camp area

(2) Ban Xieng Wang, site of documented Schistosomiasis transmission

(3) Dooley Hospital area

(4) Ban Na area

**Fig 12. Snail species.** [4]

habitats outlined in the Before Dam descriptions remain in place at Khong Island. A follow-up detailed study of the Khong Island site could provide new information to update the PBCM. The PBCM will benefit from additional updates that provide information on current environmental trends, particularly the impacts of climate change and dam construction and operation.

**Models characterizing general impacts from climate change and dams in the study area.** Models describing climate change impacts on the LMR can be useful in updating information in the PBCM. Some model-based projections of flood risks in the Lancang-Mekong River continuum predict more frequent and bigger flood events in the future in response to climate change. Through reservoir regulation, flood risks can be mitigated particularly in the upper basin where large reservoirs are located. Models indicate that the combined effects of climate change and reservoir regulation can reduce risks in the upper basin, but the increased flood magnitude and frequency, from climate change, are still dominant in the lower basin [8]. Recent reports from the Sixth phase of Coupled Inter-comparison Project (CMIP6 models), including the Standardized Precipitation Index (SPI) and Standardized Precipitation-Evapotranspiration Index (SPEI) data, predict that seasonal impacts of droughts in the Lower Mekong River Basin (LMRB) will include wetter rainy season and drier non-rainy season conditions. The SPEI shows that the middle reaches of the LMR will experience the highest levels of drought duration and intensity [27].

Reservoirs across the Lancang River reduced the annual average flow by 5% at Chiang Saen during 2008–2016, but their impacts on flow were undetectable downstream at Vientiane Laos. This flow data highlights the importance of site-specific descriptions like that provided by the PBCM since the After Dams data was collected during this period. The streamflow changes noted downstream of Mukdahan in southern Laos, Cambodia, and Vietnam were attributed to local reservoirs and climate change, with the latter changing the magnitude and frequency of floods by up to 14% in Cambodia and 45% in South Vietnam [28]. Current data on annual surface water flow in the LMR indicate a general downward trend during 1960–2010, with no clear trend after 2010. However, while most studies indicate a decreasing trend, a few studies reported the opposite with increasing flow; this may be the result of different data and methods applied in each study [28].

Hydropower development, irrigation expansion, and global climate change were examined in a study of the LMR [29]. The authors suggested studies, including transboundary collaboration in order to address sustainable development goals, could help to meet planned sustainability objectives. Other studies [30,31], using the year 2040 development scenarios, showed that hydropower and irrigation development predicted limited impacts on annual flows compared to the early development period. Predicted climate change can reduce the streamflow in June between years 2040 to 2080 for some scenarios and increase the streamflow in October in the same period. Furthermore, the streamflow showed significant increases in the long-term during September-November and December-February but showed decreases during March-May and June-August. Hydropower decreased the streamflow in the wet season by 7% but increased in the dry season by 29%. The combined impacts of all driving factors caused substantial flow reductions during the first half of the wet season by—42%, but increased the flow in the dry season by 40% and 14% compared to the early development and current scenario, respectively [29].

A recent study [32] described an integrated approach with mixed methods and assessed future sustainability challenges in social, hydrological, and ecological dimensions using a case study from the Lower Mekong basin. The study area included the combined basin of the Se Kong, Se San, and Sre Pok (3S) Rivers that deliver approximately 20% of the flow to the Mekong River system. The results suggested that future river flows in the 3S river system could

move closer to natural conditions during the dry season, with increased floods during the wet season, and require a shift in the current dam operations, from maintaining minimum flows to reducing flood hazards [32].

**Potential impacts specific to *N. aperta/S. mekongi* habitat.** The projected changes in flow in the Lancang- Mekong continuum will lead to major impacts on sediment transport and characteristics along different river channel areas and dam reservoir areas. The changes are both local and site-specific. The proposed PBCM framework can help characterize major determinants of the sediment architecture associated with the observed favorable habitat for *N. aperta* (Figs 3 and 10–12).Snail habitat sites along the lower Mekong River can be described as patchy or mosaics habitats that display high intra-and inter-site variability in substrate type, nutrient availability, and primary producer organisms that regulate food sources and water quality [33]. This was evident in the 2008–2010 studies completed by [2,3].

Sediment architecture and water quality are major determinants of the aquatic food web components supporting *N. aperta* populations, particularly macrophytes, microorganisms, and macroinvertebrates. The habitat for *N. aperta* on Khong Island described in the PBCM is conditioned by sediment architecture and particle size distribution. Sediment is a strong determinant of nutrient type, quantity, and availability to support the primary producer base of the food web. Sediments and other substrates (4) are an important source of food and attachment for snails including *N. aperta* (Figs 3 and 6–10).6Muntz [34] studied the decomposition of organic matter above and below dams in the USA. The results can be used to better understand dam impacts using the PBCM. Muntz found decomposition rates were significantly greater in the impoundments than in one free-flowing section of the river but did not find a significant relationship between decomposition and oxygen or temperature. The average temperature was slightly warmer in the impoundments, suggesting a stronger effect of temperature than oxygen on decomposition.

Current studies in China indicate similar general results with pre- and post-impoundment studies of hydrological impacts on sediment transport in the Han River Basin in China that produce a significant quantity of sediment loss downstream [35]. Sediment descriptions in the PCBM indicate a general decrease in sediment in the Khong Island, Laos area compared to upstream at Ban Dan, Thailand. Estimates of sediment transport in the Lancang-Mekong River system are highly variable but reflect conditions prevailing prior to the ongoing construction of a cascade of seven dams on the Lancang, that will trap an estimated 83% of the Lancang basin sediment when complete [36]. Thus, almost half of the total natural sediment load of the Mekong will be lost in the reservoirs of the Lancang, and the sediment load of the Lower Mekong River will consist largely of sediment derived from sources within the Lower Mekong River basin itself. These sediment losses could lead to site-specific spates of hungry water [37] in areas described in the PBCM causing habitat changes for *N. aperta* due to scouring and bank and channel erosion [38]. The important role of paleogeography and paleoclimate on speciation and endemism in isolated populations of Schistosoma vectors of the genus *Bulinus* in Africa has been described [39]. Snail vectors can be sensitive to changes in the ecological communities in which they are embedded. Macrophyte communities provide attachment substrate and nutrients for snails in littoral habitats of the LMR as described in the PBCM (Figs 11 and 12). Hydro-morphological alterations influenced the functional diversity and composition of macrophyte communities in the Danube River [40]. Macrophytes were reported to modulate the detritus cycle and change the nutrient supply in *S. mansoni* snail vector habitats [41]. Qualitative and quantitative changes in nutrients can result in significant changes in habitat structure. The enhanced release, export, and transport of diffuse nutrients from watershed forest litter in the Lancang River basin, during climate change, have been reported using remote sensing data, experimental parameters, and the SWAT model [42].

Tomczyk [43] compared the points closest to and downstream of a Hydro Powerplant (HP) in Poland and noted an increase in concentrations of most heavy metals. Although data on heavy metal contaminants were not included in the PBCM, future studies should include habitat contamination in the areas with water quality degradation. Fine-grained fractions dominate the clays and sandy clays upstream of the hydropower plants, while sands, sandy clay loams, or sandy clays are dominated downstream. The dominance of fine-grained fractions upstream of the HPs favors the accumulation of metals due to the high sorption capacity of these fractions [43]. When comparing the analyzed groups of points, the average pH values are arranged in the following sequence: upstream HP (U) > downstream HP (D). EC has opposite values than pH (i.e., U < D) EC is greater downstream. In the 1972 pre-impoundment studies, the same trend in pH and EC was noted between Ban Dan and Khong Island. The Tomcyzk study may support sustainable sediment management options for use with the PBCM and allows for the development of recommendations related to the rational management of HPs.

Shi [44] developed a niche model using machine learning algorithms to predict the probability of suitable snail habitats for the *S. japonicum* vector *Oncomelania hupensis* in China. The six greatest contributors to predicting potential *O. hupensis* distribution included silt content (13.13%), clay content (10.21%), population density (8.16%), annual accumulated temperatures of $\geq 0°C$ (8.12%), night-time lights (7.67%), and average annual precipitation (7.23%). The Before Dams description in the PBCM shows somewhat similar trends in habitat for *N. aperta* regarding the percent sand, silt, and clay at Ban Dan and Khong Island (Fig 10).

Reports describing the Lower Mekong Basin indicate that overall water quality was good up into the 2000s [1,4]. Recent studies indicate that localized degradation has occurred over the past decade near Vientiane City; the Sekong, Sesan, and Srepok Rivers, the Tonle Sap Lake system, and the Mekong Delta [45,46]. Water quality degradation likely corresponds to flow alteration, erosion, sediment trapping, and point and nonpoint wastewater. Habitat updates in the PBCM can be clarified using this type of data at specific sites.

A study of the relationship between climatic, hydrological, and water quality parameters of the LMR flowing through four different countries (Thailand, Cambodia, Lao PDR, and Vietnam) indicated that DO, pH, conductivity, Ca, Mg, Na, K, alkalinity, Cl, $SO_4^{2-}$ and Si had fair to strong negative correlations with all hydrological parameters. TSS, alkalinity, and conductivity were proposed as sensitive water quality parameters for monitoring the impacts of changing climate in the lower Mekong River [47]. Descriptions of these water quality parameters in the Before and After Dams information in the PBCM can be useful in future monitoring programs.

Current reports describing the overall mainstream water quality and ion analyses in the Lower Mekong basin near Khong Island cite good to excellent conditions like the monsoon cycle and diel cycle data reported in the PBCM Before Dams studies of 1972–73 (Fig 1–10)]; however, site-specific data were not collected as part of these surveys and other reports [45,46] cite water quality and habitat deterioration at some sites near Khong Island. A river study in France reported that dams did not disturb the seasonality of major anions but did modify silica and phosphorus concentration-flow relationships, especially during low flow. Such changes in the dynamics of river-water composition may affect downstream biological communities [48] and may occur in the LMR.

Three types of Green House Gases (GHGs), carbon dioxide, methane, and nitrous oxide emitted from reservoirs have an important but often ignored impact on climate change and need to be studied in detail in the LMRB. The biogeochemical cycle of carbon in the aquatic environment is understood to be the main mechanism for generating reservoir-based GHG emissions [49]. Future updates of the PBCM should include current data on GHGs.

The impact of temperature change on infections of *S. japonicum* in China was studied using a growing degree-days simulation model. The results predicted changes in transmission patterns and altered frequency and transmission dynamics of *S. japonicum* with climate change [50].

Kalinda [51] used a synthesis of 19 published articles and indicated that temperature rise may alter the distribution, and optimal conditions for breeding, growth, and survival of intermediate host snails, which may eventually increase the spread and/or transmission of schistosomiasis. Other studies in rivers and lakes have reported that transmission may be high at 15–19˚C. and 20-25˚C. respectively [52]. Both temperature effects studies can be used to enhance the habitat descriptions over time in the PBCM. Liu and Xu [53] studied a unique form of flow regulation (e-flow) called hydropeaking that introduces frequent, short-duration, artificial flow events to river systems in China. Hydropeaking is one form of the sporadic flows affecting habitat structure in the After Dams effects noted in the PBCM. They reported that hydropeaking drove plant species distribution, assemblage structure, and species richness-biomass relationships. Hydropeaking also generated complex species-specific effects on plant biomass patterns. They concluded that hydropeaking is likely a major driver of plant community assembly in hydropeaking-affected rivers.

Twenty sites were sampled along the length of the Mekong River from northern Thailand to Lao PDR, Cambodia, and southern Vietnam. Sites included lower Mekong River shorelines and major tributaries. Data were collected on four groups of organisms considered suitable for biomonitoring studies: benthic diatoms, zooplankton, littoral macroinvertebrates, and benthic macroinvertebrates [5,6]. Data from selected lower Mekong River sites in Laos, Thailand, and Cambodia are included in the PBCM, and results indicate that all groups were diverse and abundant in the Mekong River system with considerable variability in abundance and composition from site to site, with the mosaic conditions as described in [15,33]. In another study, thirty-two sites were assessed and classified as to their 'ecological health' using indicator characteristics, including physical and chemical measures, diatoms, and littoral macroinvertebrates. These sites were then classified as excellent, good, moderate, or poor. Ten sites were moderate and one site was poor, with the remaining sites classified as good to excellent [6].

Limpanont et al. [54] reported a new habitat site for the γ- strain of *N. aperta* (with molecular confirmation of identity) from Nong Khai Province, where it occurred in a habitat novel for this species—under paving slabs instead of under natural bed rocks as noted in [4] and PBCM data descriptions. The new site had algal aufwuchs and was located on the islet in the middle of the Mekong River. The new location is approximately four hundred km upstream from the nearest previously known site for this species.

## Discussion

Suggested variables and habitat models that can be used along with the PBCM are currently available or under development. Water management and economic development plans should fully consider e-flow recommendations that include specific criteria linked to ecological functions and processes [55,56]. Some of those functions and processes are available using descriptions from the PBCM. A robust approach that includes specific combinations of biological, physical, and chemical data could provide a useful tool to help manage river flows along different impounded areas of the Lancang-Mekong River. Assessments of e-flow should be appropriate for the context and as simple as possible. E-flow regimes should be spatially and temporally specific, connected to ecological processes and functions, and adaptive to changing conditions. One good way of accomplishing this is to develop habitat models that incorporate desired e-flow regime characteristics in the model design. Yarnell and Thoms [57] propose the

application of Functional Flows using a flow-chain model and provide two case study examples from Australia and the United States, where improvements in channel habitat and reconnection with the floodplain help achieve the desired functionality of environmental flows. A recent case study reported the use of e-flow models coupled with e-flow allocations for a tropical reservoir system by integration of water quantity (SWAT) and quality (GEFC, QUAL2K) models [58]. The coupled models allowed control of DO and other important water quality parameters that could influence snail habitat and distribution. Coupled e-flow- ecological niche habitat models could be used with information from the PBCM to provide a powerful tool for estimating the potential habitat areas that exist in the highly diverse shoreline mosaic of the Lancang-Mekong watershed. It may be possible to create predictive models for future use by describing suitable habitats for *N. aperta* and other *Schistosoma sp.* snail intermediate hosts along shoreline areas of the Lancang- Mekong River continuum. These models could be powerful management tools that combine e-flow and other habitat models describing water quality with schistosome biomarkers currently under development [59] in a One Health context to predict possible areas suitable for disease transmission.

Hydro-ecological modeling in SRH-2D and CASiMiR, utilizing flow velocity measurements and macroinvertebrate sampling data, can be considered as one method to design flow regimes that protect biological communities and should include hydrological data reflecting human influences and climate and their effects on sediment distribution [60,61].

An example of a coupled e-flow-habitat model for use in hydraulic schistosomiasis control could combine the ecological niche model as described by Shi [44], with a new model for predicting the terminal settling velocity and drag coefficient of *Oncomelania* [62]. That model could then be integrated with observations from the PBCM and current e-flow studies and habitat models [63,64,65]. Another approach could involve coupling habitat suitability models (HSMs) with frameworks for e-flow estimation models. Szalkiewicz [66] reported a comparative assessment of e-flow values for hydrobiid snails and other macroinvertebrates at two reaches of a river with different levels of hydro morphological change. They noted strong dependence of e-flows on morphological conditions and that the lower limits of e-flows noted were greater than values based on hydrology. Coupling the HSM model with data from the PBCM, including information from data-poor river systems, [16] could enhance the value of both approaches and provide a useful predictive tool for describing ecological habitats in rivers with limited data.

## Conclusions

### Value and potential applications of the PBCM framework

The PBCM model can serve as a template to summarize, compare, and combine other models describing the habitat-supporting transmission of S. *mekongi* transmission by *N. aperta*. The general model can also be used to develop other PBCMs describing other river systems and parasites. The models can help to select/predict which shoreline areas along the Mekong River could support *N. aperta* and other vector snails, as well as habitats that will support and protect the biodiversity of the riverine molluscan community in general. Niche models and habitat suitability models can be powerful tools when combined with current e-flow models. Together, the models can help to identify effective e-flow strategies to help with the control and management of waterborne disease transmission. Luo [67] completed a recent systematic review and geospatial analysis of schistosomiasis in Southeast Asia and reported that although the prevalence has been substantially reduced since 2000, focal areas with high prevalence remain. They identified areas, including Khong Island, Laos, that should be targeted for interventions to reduce disease. Models could be built using general descriptions of the littoral zone in [6,15].

PBCM models of this type have been described and suggested for other data-deficient rivers [15,16].

It may be possible to create a suite of predictive models describing suitable habitats for *N. aperta* and other *Schistosoma sp.* that are intermediate hosts along the Lancang- Mekong River continuum shoreline areas. These models could be powerful management tools, in a One Health context, by combining e-flow models with key water quality parameters and schistosome biomarkers currently under development. A Maximum Entrophy (Maxent) model predicts suitable habitats for *S. haematobium* and *S. mansoni* in Africa and is a robust snail suitability model useful in developing control and management programs for schistosomiasis in the study area [68]. Maxent models differ from the PBCM described here by requiring larger sample sizes with a minimum of 10 sampling points for reliability [69] and by requiring presence-only data [70]. Unlike the PBCM, the Maxent model only describes sites where snails are present and cannot be used to predict potential future sites capable of supporting snail intermediate hosts. *S. haematobium* and *S. mansoni* intermediate hosts are amphibian snails with a different niche than the totally aquatic *N. aperta*, and the PBCM, using both presence and absence data describing suitable habitat, may prove to be as robust and useful for sites with limited data. A Bayesian network model (sBN) delineated exposure areas of *S. japonicum* and reports high values of probability of exposure which corresponds to polygons where snails could potentially be present. The PBCM could be the basis to develop a similar model for *S. mekongi*. The schistosomiasis exposure sBN could be used by local disease control teams to identify areas of exposure and improve the efficiency of mass drug administration [71]. A study to address how snail hosts and their interaction with *S. mansoni* influence model predictions indicates that model outputs, such as schistosome prevalence in human and snail populations, respond to the inclusion of snail age structure [72]. Another study linked quantitatively hydrological drivers to distinct population dynamics through specific density feedbacks and shows that model averaging based on statistical methods yields reliable projections of snail abundance [73]. The importance of hydrological drivers of habitat structure is evident in the PBCM and snail habitat models in general. A recent study of projected changes in annual extreme rainfall, and high and low streamflow events over Southeast Asia, under extreme climate change, indicates dramatic fluctuations as predicted by a High-resolution Model Intercomparison Project (HighResMIP) multi-model experiment for the period 1971–2050 [74]. Some of the model predictions correspond with areas near the 1972 studies in Laos described in this paper and could influence future habitat dynamics of *N. aperta/S. mekongi*.

## Supporting information

**S1 Data. Raw Data for sediment collected at Khong Island, Laos and Ban Dan, Thailand, 1972.**
(XLSX)

## Acknowledgments

We wish to thank the Interlibrary Loan Staff at the University of Massachusetts, Amherst (ILLiad) for help in accessing information, and Moira Clingman for preparing the graphics.

## Author Contributions

**Conceptualization:** Guy R. Lanza, Suchart Upatham, Ang Chen.

**Data curation:** Guy R. Lanza, Suchart Upatham.

**Writing – original draft:** Guy R. Lanza.

**Writing – review & editing:** Guy R. Lanza, Suchart Upatham.

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
