## [Decision Letter · Decision Letter 0]

9 Mar 2023

Dear Professor Lanza,

Thank you very much for submitting your manuscript "A Place-Based Conceptual Model (PBCM) of Neotricula aperta Habitat During Transmission of Schistosoma mekongi on Khong Island, Laos." for consideration at PLOS Neglected Tropical Diseases. As with all papers reviewed by the journal, your manuscript was reviewed by members of the editorial board and by several independent reviewers. In light of the reviews (below this email), we would like to invite the resubmission of a significantly-revised version that takes into account the reviewers' comments. 

While the reviewers thought the study was interesting, the manuscript requires substantial editing before it is suitable for publication. Please address the comments raised by the reviewers, specifically relating to the organisation of the manuscript and the details included in the discussion/conclusion.

We cannot make any decision about publication until we have seen the revised manuscript and your response to the reviewers' comments. Your revised manuscript is also likely to be sent to reviewers for further evaluation.

Sincerely,

Krystyna Cwiklinski, PhD

Academic Editor

Alvaro Acosta-Serrano

Section Editor

While the reviewers thought the study was interesting, the manuscript requires substantial editing before it is suitable for publication. Please address the comments raised by the reviewers, specifically relating to the organisation of the manuscript and the details included in the discussion/conclusion.

Reviewer's Responses to Questions

**Key Review Criteria Required for Acceptance?**

**Methods**

-Are the objectives of the study clearly articulated with a clear testable hypothesis stated?

-Is the study design appropriate to address the stated objectives?

-Is the population clearly described and appropriate for the hypothesis being tested?

-Is the sample size sufficient to ensure adequate power to address the hypothesis being tested?

-Were correct statistical analysis used to support conclusions?

-Are there concerns about ethical or regulatory requirements being met?

Reviewer #1: Yes

Reviewer #2: • More explanation to major components of a PLACE-BASED CONCEPTUAL MODEL (PBCM) is needed.

Reviewer #3: Most of these questions are not really relevant; the manuscript is a kind of review presenting historical data on habitat charaterization of the gastropod Neotricula aperta, the intermediate host for Schistosoma mekongi, in the Mekong River prior to construction of dams and the exploring how those can be useful for predicting distribution of this snail species and transmission of S. mekongi now that many dams have been built on this river.

Reviewer #4: The objectives of the study are not clearly concentrated. While they are indeed stated throughout the manuscript, they are not upfront and clear. This paper is a mix of a long literature review, with very little of the PBCM used/discussed. Moreover, the actual structure of the PBCM is missing. Figure 6 is not adequate for discussing this, as the model is a core component of the manuscript. The methods section merely references two papers describing the methods, but more details need to be included here.

**Results**

-Does the analysis presented match the analysis plan?

-Are the results clearly and completely presented?

-Are the figures (Tables, Images) of sufficient quality for clarity?

Reviewer #1: Yes

Reviewer #2: the analysis presented match the analysis plan

the results clearly and completely presented

Reviewer #3: Results are clearly presented and Images and Tables are clear and illustrative.

Reviewer #4: While the PBCM can enhance the literature review and updated conditions, it just isn’t integrated well throughout the manuscript. For example, it would be better if the authors integrated it into more of the review/modeling scenarios starting around line 131. Furthermore, much of this section (which I presume is the Results section) needs to have some of it moved to the introduction instead. Lines 111 – 131 should be in the intro, as it motivates a lot of the study.

**Conclusions**

-Are the conclusions supported by the data presented?

-Are the limitations of analysis clearly described?

-Do the authors discuss how these data can be helpful to advance our understanding of the topic under study?

-Is public health relevance addressed?

Reviewer #1: Yes

Reviewer #2: Compare between using maxent model and a PLACE-BASED CONCEPTUAL MODEL (PBCM) in similar studies.

Reviewer #3: Conclusions are supported by data and the modelling suggested could play an important role for public health in the area.

Reviewer #4: While I think the manuscript needs significant reorganization, I do believe that the manuscript has valuable conclusions and can be improved to be publication worthy. I would suggest the authors stop including new review material in the conclusions, such as the information beginning on line 384. There is far too much new information and it feels like an aside.

**Editorial and Data Presentation Modifications?**

Reviewer #1: I have marked changes on the ms. Sci. names cannot be used as adjectives they are proper nouns

Reviewer #2: (No Response)

Reviewer #3: I have only very minor comments.

Several species or genera names are not italicized (eg lines 46, 211, 236, 331, 343 and most in the reference list)

Line 305: agal should be algal

Sentence line 291-292: add reference. 

Sometimes authors use “Schistosoma vectors” and sometimes “Schistosome vector”; Schistosoma is a genus name and should be in italics, while Schistosome is not and it should not be capitalized (see eg line 377-380).

End of line 405: L anza should be Lanza

Figure 6, left box in the third row of boxes: “Figures 1-6” should be “Figures 1-4” ?

Reviewer #4: Most of my suggestions are major suggestions, not minor. I have coalesced them below.

**Summary and General Comments**

Reviewer #1: The authors might want to consider a mention of the changes is naturally occurring fish predators of the snails with changing conditions. Just a suggestion.

Reviewer #2: (No Response)

Reviewer #3: This manuscript presents historical data on habitat characteristics for Neotricula aperta during transmission of Schistosoma mekongi in the Mekong River at Khong Island prior to dam projects. The manuscript then presents a detailed review of the effects of dams on various aspects of the river ecology and the discussion address the potential application of the Place-Based Conceptual Model (PBCM). The manuscript is well-written and addresses an interesting approach to model ecological data.

Reviewer #4: a. Overall, I believe this manuscript has valuable information, and I would like to see it published in the future. The authors have done an excellent job of combing through the literature and combining it with historical data to speak on the environmental conditions and consequences of the area. However, the manuscript suffers from an organization problem, in my opinion. Far too much of motivating information is not in the intro, the PBCM model is not described enough, and too much details in the intro and discussion that are irrelevant to the points the authors are making. For example, the introduction does not set up the reader to understand the value of the PBCM or the current conditions. It would be good if Lines 68 – 72 and 111 – 131 be moved to the intro, for example.

b. The review portion of the article needs to be cut and focused significantly. It goes on far too long. I would suggest the authors ask themselves if individual paragraphs support the overall goal of the paper, or if they are including information just to include it. If the authors do not want to get rid of these details, consider instead making an expanded review, with the focus being the review, not the PBCM. For example, it is unclear to me if the reader even needs to know any of the information in lines 155 – 168.

PLOS authors have the option to publish the peer review history of their article (what does this mean?). If published, this will include your full peer review and any attached files.

Reviewer #1: Yes: Jay R. Stauffer Jr.

Reviewer #2: Yes: mohamed kamel

Reviewer #3: No

Reviewer #4: No
---

## [Decision Letter · Decision Letter 1]

18 Aug 2023

Dear Professor Lanza,

Thank you very much for submitting your manuscript "A Place-Based Conceptual Model (PBCM) of Neotricula aperta /Schistosoma mekongi habitat before and after dam construction in the Lower Mekong River." for consideration at PLOS Neglected Tropical Diseases. As with all papers reviewed by the journal, your manuscript was reviewed by members of the editorial board and by several independent reviewers. The reviewers appreciated the attention to an important topic. Based on the reviews, we are likely to accept this manuscript for publication, providing that you modify the manuscript according to the review recommendations. 

The authors have addressed the comments raised by the reviewers. A few minor corrections are now required before the manuscript is suitable for publication.

Sincerely,

Krystyna Cwiklinski, PhD

Academic Editor

Álvaro Acosta-Serrano

Section Editor

The authors have addressed the comments raised by the reviewers. A few minor corrections are now required before the manuscript is suitable for publication.

Reviewer's Responses to Questions

**Key Review Criteria Required for Acceptance?**

**Methods**

-Are the objectives of the study clearly articulated with a clear testable hypothesis stated?

-Is the study design appropriate to address the stated objectives?

-Is the population clearly described and appropriate for the hypothesis being tested?

-Is the sample size sufficient to ensure adequate power to address the hypothesis being tested?

-Were correct statistical analysis used to support conclusions?

-Are there concerns about ethical or regulatory requirements being met?

Reviewer #1: Paper meets all of the above criteria and should be published with minor revisions.

Reviewer #3: Yes

**Results**

-Does the analysis presented match the analysis plan?

-Are the results clearly and completely presented?

-Are the figures (Tables, Images) of sufficient quality for clarity?

Reviewer #1: yes

Reviewer #3: Yes

**Conclusions**

-Are the conclusions supported by the data presented?

-Are the limitations of analysis clearly described?

-Do the authors discuss how these data can be helpful to advance our understanding of the topic under study?

-Is public health relevance addressed?

Reviewer #1: Yes -- may emphasize how and why these models should be applied to other sysrtems.

Reviewer #3: Yes

**Editorial and Data Presentation Modifications?**

Reviewer #1: A few minor corrections needed. Many are highlighted in attached ms. Scientific names cannot be used as adjectives -- they are proper nouns.

Reviewer #3: Much improved

**Summary and General Comments**

Reviewer #1: Excellent study. May want to emphasiz its usefullness to other species and systems.

Reviewer #3: I have carefully read the manuscript and have no further comments. I only found a few minor typos which probably will be detected during the type setting.

Line 24: Schistosoma should be in italics.

Line 55: Schistosome should be schistosome

Line 165: N. aperta should be in italics

Line 203: Stenothra should be Stenothyra and sp. should not be in italics

Line 394: agal should be algal

PLOS authors have the option to publish the peer review history of their article (what does this mean?). If published, this will include your full peer review and any attached files.

Reviewer #1: Yes: Jay R Stauffer, Jr.

Reviewer #3: No

Figure Files:

Data Requirements:

Reproducibility:

References

---

## [Editor Report · Decision Letter 2]

18 Sep 2023

Dear Professor Lanza,

We are pleased to inform you that your manuscript 'A Place-Based Conceptual Model (PBCM) of Neotricula aperta /Schistosoma mekongi habitat before and after dam construction in the Lower Mekong River.' has been provisionally accepted for publication in PLOS Neglected Tropical Diseases.

Best regards,

Krystyna Cwiklinski, PhD

Academic Editor

Álvaro Acosta-Serrano

Section Editor

The authors have addressed the comments raised by the reviewers. The manuscript is now suitable for publication in PLoS NTD.

---

## [Editor Report · Acceptance letter]

2 Oct 2023

Dear Professor Lanza,

We are delighted to inform you that your manuscript, "A Place-Based Conceptual Model (PBCM) of Neotricula aperta /Schistosoma mekongi habitat before and after dam construction in the Lower Mekong River.," has been formally accepted for publication in PLOS Neglected Tropical Diseases.

Best regards,

Shaden Kamhawi

co-Editor-in-Chief

Paul Brindley

co-Editor-in-Chief
